# Evaluation of Nanofiltration Membranes for Pure Lactic Acid Permeability

**DOI:** 10.3390/membranes12030302

**Published:** 2022-03-08

**Authors:** Mayuki Cabrera-González, Amal Ahmed, Khaled Maamo, Mohammad Salem, Christian Jordan, Michael Harasek

**Affiliations:** Institute of Chemical, Environmental & Bioscience Engineering E166, Technische Universität Wien, 1060 Vienna, Austria; e11811857@student.tuwien.ac.at (K.M.); e11728321@student.tuwien.ac.at (M.S.); christian.jordan@tuwien.ac.at (C.J.)

**Keywords:** lactic acid, nanofiltration, membranes

## Abstract

Lactic acid (LA) is an organic acid produced by fermentation or chemical synthesis. It plays a crucial role in the pharmaceutical, food and plastic industries. In the fermentation of, for example, grass silage, LA and different compounds are produced. To purify lactic acid, researchers have tried to investigate membrane technology to achieve a high yield of lactic acid permeance. This study tested four commercially available nanofiltration membranes (NF270, MPF-36, Toray NF, and Alfa Laval NF). Nanofiltration experiments were performed to investigate the rejection levels of lactic acid from a binary solution by using distinct molecular weight cut off membranes. All of the experiments were conducted with a lab-scale cross-flow membrane unit. Different operating conditions (pH, temperature) were studied for each membrane; the optimal process condition was found at 25 °C and pH 2.8. With higher temperatures and pH, an increase in LA rejection was observed. The MPF-36 membrane shows the lowest lactic acid rejection yield of 7%, while NF270 has the highest rejection yield of 71% at 25 °C and pH 2.8. These results will be helpful in the future to understand both the interaction of lactic acid permeance through nanofiltration membranes and process scale-up.

## 1. Introduction

Lactic acid (LA) is an essential chemical compound used as a flavour, acidifier, and preservative in the food industry. The pharmaceutical, cosmetic and polymers industries use lactic acid as a raw material to develop commercial products [1]. Lactic acid is produced in two ways: fermentation (biotechnological process) and chemical synthesis. Different feedstocks have been utilised for lactic acid production to replace the oil-based material. For example, green biomass, like grass or seaweed, can be fermented to obtain lactic acid [2]. The fermentation process for lactic acid production is performed by lactic acid bacteria (LAB) through metabolic pathways. LAB, such as *Lactobacillus delbrueckii* [3] or *Bacillus coagulants* strains A20, A369, A107, and A59 [4], can convert sugars like fructose, glucose, arabinose, etc., into lactic acid, ethanol, butyric, propionic, acetic, and caproic acid, and other organic acids [5]. Although fermentation has many advantages, the production of other chemical compounds besides lactic acid is not desirable in the industries mentioned above, since they require a pure form of this compound (LA) [6]. Therefore, LA without impurities or in a highly refined form is mandatory for industrial application [7].

Even though the molecular compounds generated in the fermentation are potentially valuable products, downstream processing steps are necessary to purify and recover lactic acid and to remove the undesired compounds.

Downstream processing (DP) is a series of unit operations that remove most of the impurities from a complex solution to obtain a set of pure chemical compounds in different stages of the entire process. All of the required steps in DP establish an expensive process to recover lactic acid from any feedstock, and it can cost between 50% [8] and 80% of the total production [9]. A conventional route for lactic acid recovery goes from fermentation to extraction, then distillation, after that adsorption to go through membrane filtration, to evaporation and to end with a crystallisation [10]. Therefore, the typical route can be replaced with selective membranes to recover lactic acid. Membrane technologies are used in the downstream processing of chemical and biological industries [11]. The advantages of membranes are that they are highly selective, have high levels of purification, can be integrated into conventional fermenters and reactors, and are flexible in the scale of production [12]

Nanofiltration (NF) is a membrane separation technique situated between ultrafiltration and reverse osmosis. The nominal molecular weight cut-off (MWCO) of NF is in the range of 200 to 1000 g mol^−1^ [13]. NF is ideal for purifying lactic acid as there is no need to use additional chemicals [14]. According to the cost–benefit and selectivity, the nanofiltration separation process is more competitive than conventional separation. In addition, the high rejection of small organic molecules and multivalent inorganic salts at modest applied pressures are some of the essential advantages of NF [15].

The importance of nanofiltration membranes technology has increased in many industries, specifically biotechnology, in the last decade. NF membranes are used in downstream processing because they separate proteins, nutrients, cells, unconverted carbon sources, and salts [16] from the fermented broth.

Several authors have addressed the separation of lactic acid from complex solutions (acid whey, sugar bread, and crust bread, among others.) [6,17,18,19]. However, there is no extensive research for nanofiltration on a binary solution of lactic acid. The study of the binary solution of lactic acid in the membrane process will allow to understand the transport phenomena of this molecular compound through nanofiltration [20].

One of the critical parameters in transport phenomena is mass transfer. This nonequilibrium process involves driving forces: electrical potential, temperature, concentration and pressure, selective sorption, mechanical sieving, and diffusion through the membranes [21].

Diffusion plays a vital role in chemical processes, such as porous catalysis, across phase interfaces and porous membranes, within fluids and gels. The diffusion coefficient is a key parameter to design mass transfer and membrane performance evaluation [22].

The Equation of Maxwell–Stefan describes the mixture transport of a binary or multi-compound solution to predict the separation performances through membranes based on the solution-diffusion model [23]. Therefore, experimental work needs to be conducted to use and support this model for simulation purposes.

This research aims to study the specific permeance of lactic acid as a binary solution in four different commercially available membranes. In addition, the effect of pH, temperature, and membrane pore size in the permeance of lactic acid were investigated. This is a preliminary experimental investigation of the retention of lactic acid in NF. The obtained data will be helpful to understand the interactions between LA and membrane properties at different pH and temperatures. In addition, to gain an overview to help choose the best membrane performance for lactic acid permeability for future downstream processing.

## 2. Materials and Methods

### 2.1. Preparation of the Lactic Acid Solution

Lactic acid (C_3_H_6_O_3_; ≥85%, Sigma-Aldrich, Wien, Austria) was used as a raw material to prepare the binary solution in this work. First, 50 g of LA were dissolved in 2 L of deionised water, at room temperature in an air atmosphere. The final concentration for C_3_H_6_O_3_ was 0.277 mol L^−1^. This concentration is based on green silage juice [24]. The pH of the initial solution was 2.8, and then it was adjusted to 3.9 and 6.0 by adding 7 and 14 g of NaOH, respectively. The pH was adjusted to approach the pH of grass silage (Zhao et al., 2021). The physicochemical properties of lactic acid are presented in Table 1.

### 2.2. Experimental Set-Up and Nanofiltration Membranes

The separation performance of NF270, MPF-36, Toray NF, and Alfa Laval NF commercially available flat sheet membranes were evaluated separately in the nanofiltration of the lactic acid solution. A lab-scale cross-flow filtration membrane unit, model OS-MC-01 (Figure 1), was used to carry out the experiments. The unit is equipped with a 2 L capacity stainless steel jacketed feed tank. The feed solution is pumped to the rectangular cross-flow membrane module, with an effective membrane area of 0.008 m^2^ (0.04 m × 0.2 m), through a CAT-high pressure piston pump model 231, with a maximum flow capacity of 3.7 L min^−1^ and a pressure up to 60 bar. All of the experiments were conducted in a batch concentration mode; the retentate was recycled back to the feed tank, and the permeate was continuously exiting the system.

The four nanofiltration membranes (Table 2) were chosen according to the different molecular weight cut-off (MWCO). To determine the permeance of LA and the performance of each membrane, samples of the permeate and the retentate were taken at the end of the nanofiltration process.

The nanofiltration process finished when there was 1400 g in the permeate, the remaining 30% of the solution was left in the feed tank to take the retentate samples for concentration analysis and to avoid the dry run of the pump. The permeance of the LA is directly related to the rejection of LA. The concentration of LA in the permeate and retentate was detected by HPLC.

Before the experiment, all the membranes were hydrated by being inserted in deionised water for 20 min before use. Then, the membranes were compacted in the module for 20 min more under pressure at 32 bar and a 3.6 L min^−1^ cross-flow rate.

### 2.3. Operating Conditions

The experiments were carried out using the solution mentioned in Section 2.1. The nanofiltration process evaluated two independent variables (pH and temperature) to determine their effect on lactic acid permeance and concentration. Table 3 presents the operating conditions which were applied for each membrane. The following parameters were calculated for each membrane: water flux at the beginning and the end of every experiment, the retention coefficients of lactic acid, and permeate flux. A VWR thermo-bath controlled the temperature during the experiments. Conductivity, permeate flux, and pH were measured every 10 min until 70% of the model solution was collected. A total of 16 experiments were carried out.

The osmotic pressure of the feed in the binary solution containing the lactic acid was 7 bar. During the batch mode, the concentration in the feed tank increases with time due to the solvent removal, which leads to the increased osmotic pressure of the solution; therefore, the applicable constant driving force at high pressure was 32 bar to avoid flux reduction. The membranes used in this study are permeable for water at high pressure. The temperature of nanofiltration was chosen according to the biomass fermentation at 40 °C for lactic acid production [28]. Regarding 25 °C, the temperature was used to compare the rejection of lactic acid at room temperature.

### 2.4. Lactic Acid Quantification

The concentration of LA in the feed, permeate, and retentate was quantified by High-Performance Liquid Chromatography (HPLC). All the samples were diluted to 1:8 to match the HPLC detection range. The method used in the HPLC is shown in Table 4.

The rejection of lactic acid was calculated from Equation (1) to determine the performance of the nanofiltration membrane. $C_{p}$ and $C_{f}$ are the concentrations in the permeate and the feed, respectively.
(1)$$R=\left(1\;-\left(\frac{C_{P}}{C_{f}}\right)\right)\times \;100\%$$

## 3. Results and Discussion

### 3.1. Water Flux and Flux Reduction

Pure water flux is one of the parameters used to describe membrane characteristics [29]. In the separation process, the feed solution affects the performance of the membranes in terms of flux. The particles or colloids interact physically with the membrane, which causes a pore or surface layer blocking [30]. Organic substances can produce flux reduction because they can be attached via adsorption in the membrane. On the other hand, inorganic compounds can also form membrane blocks because they can precipitate dissolved components due to oxidation or pH variation [30]. Figure 2 presents the pure water flux obtained from the four nanofiltration membranes at different experimental conditions (Table 3). The following Equation (2) calculates the flux reduction:(2)$${\mathrm{FR}}_{\mathrm{PWF}}\left(\%\right)=\frac{{\mathrm{PWF}}_{b}{\;-\;\mathrm{PWF}}_{a}}{{\mathrm{PWF}}_{b}}\;\times \;100\%\;$$

FR_PWF_ is the flux reduction in the pure water flux%, PWF_a_ is the pure water flux after the LA filtration in Kg m^−2^ h^−1^, and PWF_b_ is the pure water flux before the LA filtration in Kg m^−2^ h^−1^ [31].

The highest water flux was for NF270, MPF-36, Toray NF, and Alfa Laval NF at 40 °C, which exhibited a flux of 520, 372, 292, and 320 Kg m^−2^ h^−1^, respectively, before the filtration of the binary solution.

After the nanofiltration of a lactic acid binary solution, no flux reduction occurred when Toray NF was used in experiment 1 (Figure 2a) and experiment 4 (Figure 2d), as well as in Alfa Laval in experiment 2 (Figure 2b), because the FR_PWF_ is 0%. In this case, it can be assumed that there is no effect on the surface of the membrane from the filtration of the LA solution. The most affected membrane with the highest flux reduction was NF270 in experiment 3 (Figure 2c) and experiment 4 (Figure 2d), with a FR_PWF_ of 15% on average.

In addition, no flux reduction was observed for MPF-36 for pH 6.0. On the other hand, the membrane NF270 had a decreasing flux at pH 3.9 and 6.0; in both cases, the water flux diminished by 15% regarding the FR_PWF_ for this membrane. Even though the NF270 presented the highest water flux compared to the other membranes used in this study, this membrane experienced the most flux reduction. Figure 3 represents a 3D image of NF270 after being tested for LA permeability, measured by a 3D laser-scanning microscope (Keyence VK-X3000 Series) to observe the change in the structure of the membrane. This finding is concordant with [32], who found the same decline in flux for NF270 compared to this study.

The water flux is higher and quick when the membrane is thin. In contrast, when a membrane is thick, the water flux is lower and slow due to the nanochannels being larger [33]. The thickness of the membrane for NF270 is 7 to 14 nm [34]. For that reason, the water flux is high due to the NF270 being a thin membrane.

### 3.2. Conductivity

The conductivity was measured in the permeate and the retentate vs. time, to evaluate the performance for lactic acid permeability in each membrane. The conductivity helps as a quick test to determine if there is a migration of ions from the feed tank to the permeate through the membrane. The conductivity was measured with a WTW TetraCon 925 conductivity probe coupled to a WTW Multi 3430. Samples from the retentate and the permeate were taken every 10 min. The measurements of the conductivity are shown in Figure 4, where the curves with the shapes filled in black represent the retentate, and the unfilled shapes represent the permeate.

The pH adjustment influenced the conductivity of the retentate directly in all of the tested membranes. The conductivity in the retentate increased by 21, 24, 85, and 91% compared to the feed (initial measurement) for experiments 1, 2, 3, and 4, respectively, using the membrane NF270 (Figure 4a). Experiment 4 for NF270 had a substantial increase in the conductivity of the retentate, from 10,370 to 19,300 µS/cm. This result can be attributed to the high pH adjusted by NaOH, which increases the presence of ions in the solution and increases the lactic acid dissociation. Moreover, the conductivity in the retentate for the SelRO^®^ MPF-36 membrane increased by 4, 22, 33, and 34% for experiments 1, 2, 3, and 4, respectively, compared to the initial feed which is represented in Figure 4b. The slightly increasing retentate conductivity regarding experiments 3 and 4 in MPF-36 is due to the MWCO of 1000 g mol^−1^; therefore, this characteristic allows NaOH and lactic acid to pass through the membrane. There was only a 1% difference in the conductivity between pH 3.9 and 6.0, and the lowest increment was in experiment 1. Regarding Toray NF (Figure 4c), the conductivity in the retentate increased by 26, 26, 74, and 91% for experiments 1, 2, 3, and 4, respectively, compared to the feed. This means that when Toray NF is used, the conductivity in the retentate at 25 °C and 40 °C increases in the same percentage at pH 2.8. Concerning the Alfa Laval membrane (Figure 4d), the conductivity in the retentate also increased compared to the feed solution by 21, 29, 100, and 86% for experiments 1, 2, 3, and 4, respectively. The Alfa Laval NF membrane showed a considerably high difference in experiment 3 in terms of the conductivity of the retentate being a double value compared to the initial feed, from 5000 to 10,000 µS/cm. In the four tested membranes, the conductivity in the permeate side was lower than in the retentate at pH 2.8. It suggests that lactic acid does not pass entirely to the permeate side.

The conductivity in the permeate and the retentate depends on the MWCO of the membrane and the charge. With a higher MWCO, the conductivity in the retentate stream is lower compared to the lowest MWCO of the membranes. This behaviour can be related to the sieving effect in the MPF-36 membrane regarding the pore size. In addition, the increase in pH is directly related to the conductivity increment because LA dissociates at a higher pH above 3.86.

### 3.3. pH Variation in the Permeate and Retentate

pH is a key factor that strongly influences the membrane and the electrolyte solution, specifically in weak acids [35], e.g., lactic acid. pH also affects the charge of the active and selective layers of membranes. The pH is measured with a WTW SenTix H pH electrode coupled to a WTW Multi 3430 pH meter, calibrated against standard buffers at pH 4.00, 7.00, and 10.00. Samples were taken every 10 min. The results of the measurements are presented in Figure 5. The curves with the shapes filled in black represent the retentate, and the curves with the unfilled shapes show the permeate.

After the membrane filtration of the lactic acid solution at pH 2.8, 3.9, and 6.0, the pH of the resulting retentate was 2.8, 4.1, and 6.1, respectively, for NF270 (Figure 5a). For the SelRO^®^ MPF-36 membrane (Figure 5b), the initial pH was 2.8, 3.9, and 6.0, and the obtained pH of the retentate after the filtration process was 2.6, 3.8, and 6.0, respectively. For the MPF-36 membrane, the pH of the retentate was slightly lower than the initial pH at 2.8 and 3.9. For the Toray NF membrane (Figure 5c), the pH variation occurred for 2.8, which decreased by 7% in the retentate compared to the feed, for the pH 3.9 in the feed, the retentate increased by 6%, with a final pH of 4.1; it shows the same behaviour as NF270. The tight membrane with a MWCO of around 200–300 g mol^−1^ has a similar tendency in pH variation over time (Figure 5d). However, a loose membrane, such as MPF-36, with a MWCO of 1000 g mol^−1^, shows no differences in the pH variation of both the permeate and the retentate over time.

### 3.4. Lactic Acid Permeability

The term permeability in this work refers to the accumulated amount of lactic acid in the permeate. The objective of the nanofiltration was to have low lactic acid rejection. This means lactic acid concentration must be low in the retentate, and the concentration of this compound should be high in the permeate.

#### 3.4.1. Effect of the pH

The pH of the solution strongly influenced lactic acid permeability (Figure 6). The retention of LA increased with the pH increasing. At pH 3.9, the retention of LA was 73% for NF270, 24% for MPF-36, 80% for Toray NF, and 80% for Alfa Laval NF.

The lowest rejection of lactic acid was at pH 2.8. For NF270 this was 71%; for MPF-36 it was 7%, for Toray NF it was 32%, and for Alfa Laval NF it was 40%.

The lowest rejection of lactic acid contained in the binary solution was for the MPF-36 membrane at pH 2.8 and 25 °C reaching a 7%, Figure 6. Therefore, the permeability of LA was 93% in the permeate.

At a high pH (above 3.86), LA is dissociated in lactate (C_3_H_5_O_3_^-^) and proton (H^+^) due to the pKa of the lactic acid. On the other hand, nanofiltration membranes have negatively or positively charged surfaces depending on the pH [36]. A membrane charged negatively will reject most lactate due to electrostatic repulsion, whereas a positively charged membrane will pass most lactate due to electrostatic attraction. The electrostatic interaction is produced in all commercial membranes due to an isoelectric point. The isoelectric point is the neutral charge of the membrane at a specific pH. The pH range for the isoelectric point among all the membranes varies depending on the composition.

Regarding the relationship between nanofiltration membranes and pH, if the solution pH is lower than the isoelectric point (IP), the membrane will be positively charged, and membranes will be negatively charged if the pH is higher than the IP. This modification of charges leads to changes in porosity and the membrane surface conformation. As a result, there is a reduction in the permeate flux.

In the case of experiments at pH 3.9 and 6.0, these are over the isoelectric point of each membrane; therefore, in every investigation, the membrane is negatively charged except for MPF-36, which was only negative at pH 6.0. It avoids lactate transport through the membranes, as lactic acid at a pH over 3.86 is dissociated, resulting in a high rejection of lactic acid. In addition, the increase in the pH of the LA solution leads to an increase in viscosity [14].

The dissociation constant of any compound is calculated by the Equation of Henderson–Hasselbalch [36]. The dissociation degree of lactic acid depends on the pH. The high yield could be obtained by changing the pH. pH 3.8 and 6.0 lead to more dissociated LA than lactate [24]. However, LA remains undissociated for the lower pH 2.8; therefore, the permeability is higher. The dissociation of lactic acid at pH 2.7 is 6.47, at pH 3.9 is 48.14, and at pH 6.0 is 99.28%, respectively [37].

At a higher pH above 3.8, the dissociation of lactic acid increases and leads to increased rejection. Therefore, it is concluded by our experimental work that pH plays a role in the transport of lactic acid through nanofiltration membranes. Kumar et al., 2020, achieved 37% lactic acid rejection at pH 2.5 through partitioning methods [6]. These results agree with our research; at lower pH, the permeability of LA is higher. The pH of the permeate and the retentate of lactic acid transport through membranes is required to validate any mathematical simulation model.

On the other hand, the flux of the binary solution of lactic acid decreased markedly for NF270, Toray NF, and Alfa Laval when the pH increased. For NF270, MPF-36, Toray NF, and Alfa Laval at pH 2.8 compared to pH 6.0, the flux decreased in all the experiments by 53%, 8%, 26%, and 33%, respectively. In addition, there is a correlation between rejection and flux. At a high rejection of lactic acid, the flux was lower; therefore, the pH plays an important role with both parameters.

#### 3.4.2. Effect of Temperature

At high temperatures, LA is dissociated in the solution [37], and the membranes allows sorption through them. The rejection of lactic acid is affected by increasing the temperature for the Toray NF and Alfa Laval NF membranes due to the Donnan effect, while for MPF-36, the Donnan and the sieving effect (Figure 7). In contrast, the rejection of lactic acid decreases at higher temperatures due to sorption for NF270.

The rejection of LA on the NF270 membrane decreased by 10% at 40 °C compared to 25 °C. On the other hand, lactic acid retention increased by 88, 51, and 21% for MPF-36, Toray NF, and Alfa Laval NF, respectively.

Temperature is one important operating parameter that improves the flux and affects the rejection of LA. LA retention increased by 88% for MPF-36, and the lowest rejection increase was 21% for Alfa Laval NF when the temperature was 40 °C in comparison with 25 °C. However, for NF270, the rejection of lactic acid decreased when the temperature increased.

The highest permeability yield (93%) for lactic acid was obtained when MPF-36 was used at 25 °C and pH 2.8. This yield is close to Novalin and Zweckmair, 2009 [38], with 89% of LA permeability.

Regarding the flux in the nanofiltration of the lactic acid binary solution, it increases when the temperature increases as well as the rejection. The flux increased by 34, 28, 35 and 50% for NF270, MPF-36, Toray NF, and Alfa Laval NF, respectively, when the temperature rises from 25 to 40 °C. The most affected membrane regarding the flux when the temperature increased was Alfa Laval due it reaches 50% of a higher flux.

## 4. Conclusions

Lactic acid is one of the leading products from grass silage juice; it has important uses in numerous industrial applications. The permeability of LA is an essential consideration in purifying LA when using the adequate membrane for downstream processing. Nanofiltration is used to study the transport of binary solutions (lactic acid and water).

The experiment showed that nanofiltration is useful for lactic acid separation but depends strongly on the characteristics of the membrane regarding the selective layer charge, the pore size, and the composition. MPF-36 presented the best performance for lactic acid permeance at 25 °C and pH 2.8, with a 93% yield; as MPF-36 is a loose membrane, the permeance is mainly due to its pore size characteristic. However, a poor performance was given by NF270 with an LA rejection of 71% at the same operating conditions. The optimal operating parameter for lactic acid transport was found at pH 2.8 and 25 °C in all four tested membranes.

On the other hand, when the pH increases, the flux decreases for the binary solution of lactic acid; in contrast, the flux increases as well when the temperature increases.

The pH variation of the feed directly influences the charge of the membrane due to the isoelectric point. Therefore, this parameter must be considered to recover certain compounds and their dissociation grades from obtaining a high permeability of lactic acid.

The experimental data of the pH and the conductivity in both the permeate and the retentate of the process will help select the optimal lactic acid permeability performance for further downstream processing and process scale-up.

## Figures and Tables

**Figure 1 membranes-12-00302-f001:**
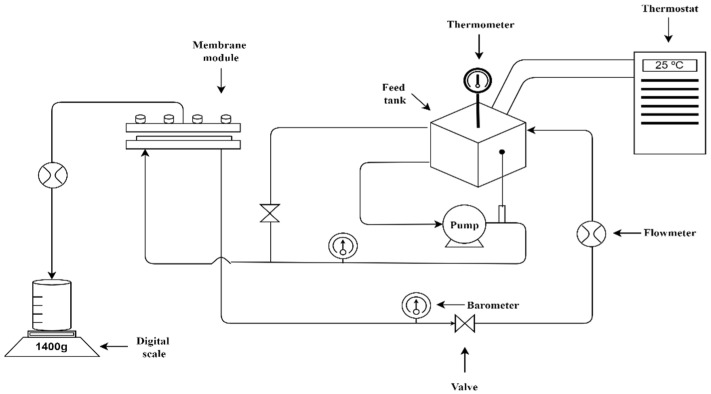
Schema of membrane test cell (model OS-MC-01).

**Figure 2 membranes-12-00302-f002:**
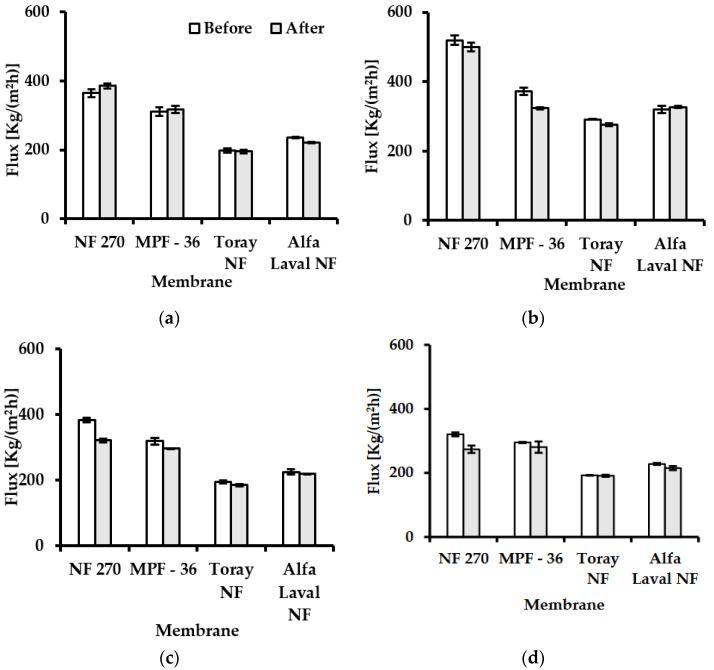
Water flux before and after the experiment with lactic acid for NF270, MPF-36, Toray NF, and Alfa Laval NF membranes. (**a**) at pH 2.8, T: 25 °C and 32 bar. (**b**) at pH 2.8, T: 40 °C and 32 bar. (**c**) at pH 3.9, T: 25 °C and 32 bar. (**d**) at pH 6.0, T: 25 °C and 32 bar.

**Figure 3 membranes-12-00302-f003:**
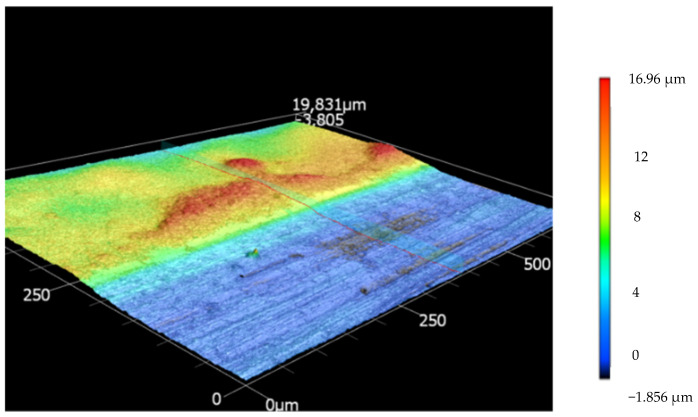
NF270 3D image. 20× measured by 3D laser-scanning microscope.

**Figure 4 membranes-12-00302-f004:**
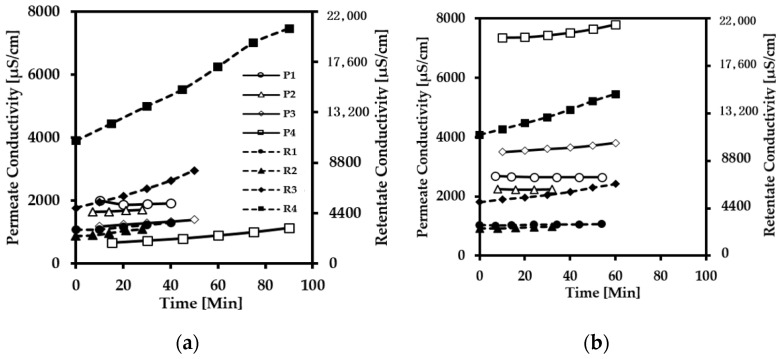
The conductivity of the lactic acid solution thought different membranes: (**a**) NF270, (**b**) SelRO^®^ MPF-36, (**c**) Toray NF, and (**d**) Alfa Laval NF. P1 and R1, P2 and R2, P3 and R3, and P4 and R4 belong to experiments 1, 2, 3, and 4 listed in Table 3. P is permeate and R is retentate.

**Figure 5 membranes-12-00302-f005:**
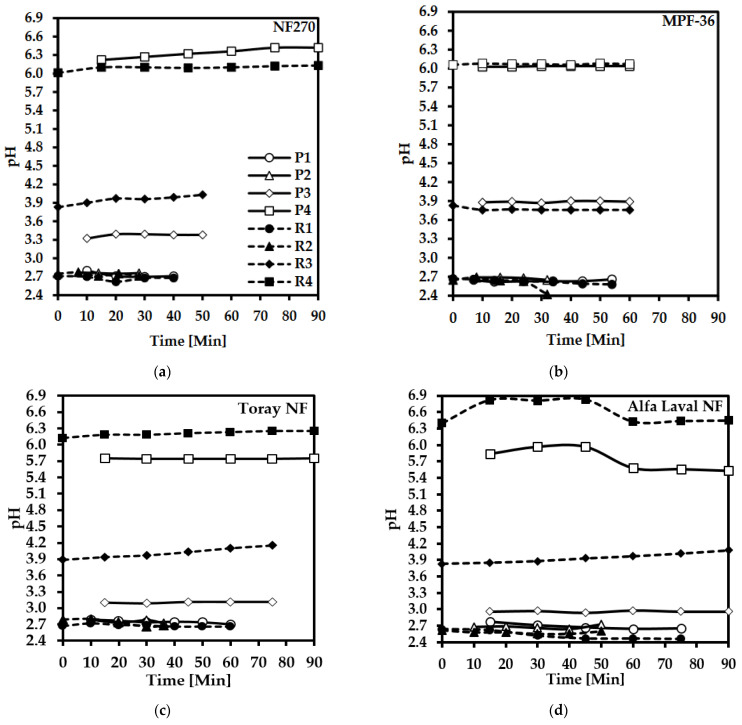
pH variation of the binary solution of lactic acid over time using different membranes: (**a**) NF270, (**b**) SelRO^®^ MPF-36, (**c**) Toray NF, and (**d**) Alfa Laval NF. P is the permeate, R is the retentate, and the numbers indicate the experiments listed in Table 3.

**Figure 6 membranes-12-00302-f006:**
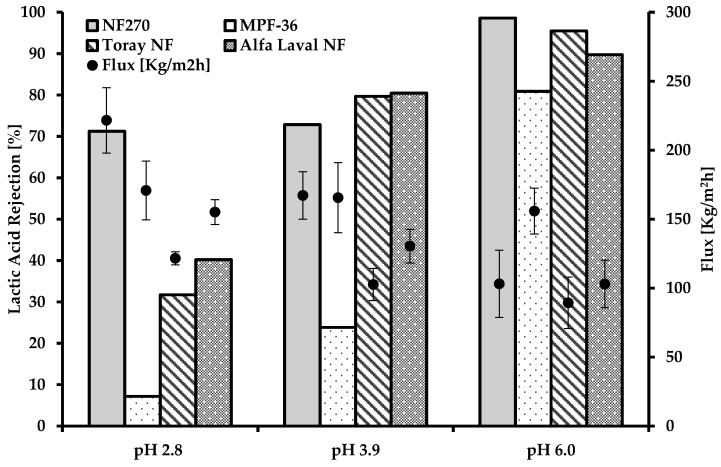
Rejection of lactic acid at different pH.

**Figure 7 membranes-12-00302-f007:**
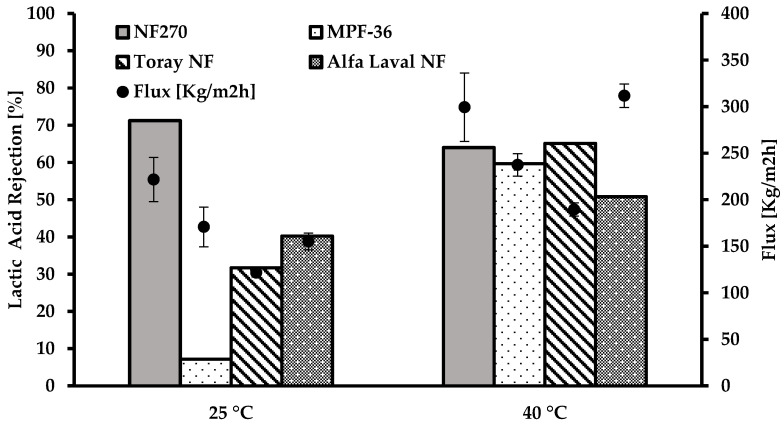
Rejection of lactic acid at different temperatures and flux variation.

**Table 1 membranes-12-00302-t001:** Physicochemical properties of the lactic acid.

Property	Value
Molecular structure	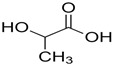
Molecular formula	C_3_H_6_O_3_
Molecular weight g mol^−1^	90.08
Dissociation constant (pKa) at 25 °C	3.86
Dissociation constant (pKa) at 40 °C	3.67
Diffusion coefficient at 30 °C [25]	11.2 × 10^−10^ m^2^ s^−1^

**Table 2 membranes-12-00302-t002:** Characteristics of the nanofiltration membranes.

Parameter	NF270	SelRO^®^ MPF-36	Toray NF	Alfa Laval NF
Manufacture	FilmTec™	Koch	Toray	Alfa Laval
Material	Polypiperazine	Polysulfone	Polypiperazineamide	Polyamide
MWCO (g mol^−1^)	200	1000	200	300
Maximum operating temperature (°C)	45	60	50	50
Operating pH range	3–10	3–10	3.5–10.5	3–10
Max operating pressure (bar)	41	35	55.2	55
Isoelectric point (pH)	3.6 [26]	5–6.5 [27]	4.0	4.0

**Table 3 membranes-12-00302-t003:** Experimental conditions for lactic acid permeability.

Experiment	Pressure (bar)	Temperature (°C)	pH	Lactic Acid (g L^−1^)
1	32	25	2.8	25
2	32	40	2.8	25
3	32	25	3.9	25
4	32	25	6.0	25

**Table 4 membranes-12-00302-t004:** The method used in the HPLC.

Experiment	Value
Equipment	Shimadzu UFLC
Flow (mL min^−1^)	0.6
Injection volume (µL)	10
Mobile phase of H_2_SO_4_ (mM)	5
Gradient	Isocratic
Oven temperature (°C)	50
Refractive index detector	RID-10A
column	Shodex SH1011 (8 × 300 mm)
Guard column	SH-G Sugar

## Data Availability

Not applicable.

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
