# Peer review of "Evaluation of Nanofiltration Membranes for Pure Lactic Acid Permeability"

_membranes, 2022, doi:10.3390/membranes12030302_

Round 1
Reviewer 1 Report
This article is about evaluation some commercial nanofiltration membranes to recovery lactic acid. The authors investigated the membranes in some experimental condition for the purpose of the LA recovery. When it comes to novelty, Nevertheless, I do not see anything new or interesting in this paper. Additionally, such an article is more like a report than a scientific paper, as there is a lack of discussions, all we see here are some experimental results without providing strong evidence and relationship between them. I believe these features are essential for papers to be published in the membranes journal.
Some specific comments:
- the article was not well prepared and inaccuracies are obvious in the content and sentences, making the readers confused.
- All the experimental results they presented can be obtained form a whole lot of previous papers published during the last 10 years. Hence, there is no new investigation in this work!
- Most of the references they used in their paper are old-fashioned! They should have done a decent literature review before starting this study.
Based on the above-mentioned comments the paper is not appropriate to be published in the membranes journal and I suggest it be rejected.
Author Response
Dear Reviewer,
Please see the attachment below.
Best regards,

Reviewer 2 Report
After reviewing this manuscript entitled: "Evaluation of Nanofiltration Membranes for Pure Lactic Acid Recovery", I found it an interesting study with respected exerted effort.
I can recommend it to be accepted for publication in (membranes) in the current form.

Author Response
Dear reviewer 2,
Please see the attachment below.
Best regards,

Reviewer 3 Report
In this manuscript, nanofiltration (NF) were performed with commercial NF membranes for lactic acid (LA) purification. Several commercial NF membranes were tested in various operating conditions (pH, temperature). The results showed that the LA separation performance can vary with both NF membrane properties and operating conditions. However, the explanations and discussion are insufficient. The manuscript must be improved for the publication.
- In Introduction, a brief review on lactic acid production from green silage juice and conventional downstream process for lactic acid recovery is needed. Also, the authors should support "NF is ideal for purifying lactic acid~" with references on actual NF application for LA separation. Also, what is the aim and the novelty of this work? The purpose and advantages of NF for LA purification or separation are unclear.
- Although the author showed the potential of NF for LA purification processes, the NF experiment were performed with only pure LA aqueous solution in this study. The purity of LA permeate and impurity removal efficiency of NF process can not be judged.
- What is the meaning of conductivity? both LA and NaOH affects on the conductivity of the solution. How can we analyse them separately?
- In overall, discussion on the results (figures) are insufficient.
- In Figure 7, what information we can get? also, how are the other membranes or conditions?
Author Response
Dear Reviewer 3,
Please see the attachment below.
Best regards,

Reviewer 4 Report
The manuscript entitled “Evaluation of Nanofiltration Membranes for Pure Lactic Acid Recovery” has been evaluated. In this work, four flat-sheet commercial NF membranes were employed to recover lactic acid under different operating conditions (pH, temperature). The work is within the scope of the journal and is of practical value. The comments are as follows:
- Since lactic acid disassociates under testing condition, therefore, the electrostatic interaction difference between the four membranes should be investigated along with the MWCO difference.
- Figure 1: the flux data of each tested membrane before filtration are much different, please explain.
- The unit of MWCO and molecular weight should be consistent.
- Figure 2: the use of “water permeability” is not correct, flux will be more reasonable.
- Figures 5 and 6: experimental errors should be provided.
- Many typo errors should be revised.
Author Response
Dear reviewer 4,
Please see the attachment below.
Best regards,

Round 2
Reviewer 3 Report
The manuscript has been sufficiently improved.
1. It would be nice to divide Fig. 6 and Fig. 7 according to the membrane or test conditions(pH and temperature).
Author Response
Dear Reviewer,
Please see the attachment.
Best regards,
Mayuki Cabrera-González.
